# Prompt as Triggers for Backdoor Attack: Examining the Vulnerability in Language Models

**Shuai Zhao[1][3], Jinming Wen[1], Luu Anh Tuan [3], Junbo Zhao [4], Jie Fu[2]***

[1] Jinan University, Guangzhou, China;
[2] Hong Kong University of Science and Technology, Hong Kong, China;
[3] Nanyang Technological University, Singapore;
[4] Zhejiang University, Zhejiang, China;

n2207879d@e.ntu.edu.sg; jinming.wen@mail.mcgill.ca; anhtuan.luu@ntu.edu.sg

j.zhao@zju.edu.cn; jiefu@ust.hk

## Abstract

The prompt-based learning paradigm, which bridges the gap between pre-training and fine-tuning, achieves state-of-the-art performance on several NLP tasks, particularly in few-shot settings. Despite being widely applied, prompt-based learning is vulnerable to backdoor attacks. Textual backdoor attacks are designed to introduce targeted vulnerabilities into models by poisoning a subset of training samples through trigger injection and label modification. However, they suffer from flaws such as abnormal natural language expressions resulting from the trigger and incorrect labeling of poisoned samples. In this study, we propose **ProAttack**, a novel and efficient method for performing clean-label backdoor attacks based on the prompt, which uses the prompt itself as a trigger. Our method does not require external triggers and ensures correct labeling of poisoned samples, improving the stealthy nature of the backdoor attack. With extensive experiments on rich-resource and few-shot text classification tasks, we empirically validate ProAttack's competitive performance in textual backdoor attacks. Notably, in the rich-resource setting, ProAttack achieves state-of-the-art attack success rates in the clean-label backdoor attack benchmark without external triggers[1].

## 1 Introduction

The prompt-based learning paradigm (Petroni et al., 2019; Lester et al., 2021; Liu et al., 2023), which utilizes large language models (LLMs) such as ChatGPT[2], LLAMA (Touvron et al., 2023), and GPT-4 (OpenAI, 2023), achieves state-of-the-art performance in natural language processing (NLP) applications, including text classification (Min et al., 2022), machine translation (Behnke et al., 2022), and summary generation (Nguyen and Luu,

2022; Zhao et al., 2022b, 2023). Although prompt-based learning achieves great success, it is criticized for its vulnerability to adversarial (Zang et al., 2020; Zhao et al., 2022a; Minh and Luu, 2022) and backdoor attacks (Wang et al., 2020; Zhou et al., 2023). Recent research (Chen and Dai, 2021; Xu et al., 2022; Cai et al., 2022) shows that backdoor attacks can be easily carried out against prompt-based learning. Therefore, studying backdoor attacks becomes essential to ensure deep learning security (Qi et al., 2021c; Li et al., 2022).

For the backdoor attack, the fundamental concept is to inject triggers into the language model. Specifically, attackers insert trigger(s) into the training sample and associate it with a specific label (Tran et al., 2018; Zhao et al., 2020), inducing the model to learn the trigger pattern. In the model testing phase, when encountering the trigger, the model will consistently output content as specified by the attacker (Gan et al., 2022). Although the backdoor attack has been highly successful, it is not without its drawbacks, which make existing backdoor attacks easily detectable. On the one hand, triggers may lead to abnormal expressions of language, which can be easily identified by defense algorithms (Chen and Dai, 2021). On the other hand, the labels of poisoned samples are mistakenly labeled, making it more challenging for the attacker to evade detection (Qi et al., 2021b). Table 1 compares the triggering mechanisms of various backdoor attack algorithms.

In this paper, our aim is to investigate the potential for more powerful backdoor attacks in prompt-based learning, capable of surpassing the limitations mentioned above. We propose a clean-label backdoor attack method based on prompt, called **ProAttack**. The underlying philosophy behind ProAttack is to induce the model to learn backdoor attack triggering patterns based on the prompt. Specifically, we engineer the poisoned samples utilizing special prompts, where the labels are cor-

---

* Corresponding author.

[1] https://github.com/shuaizhao95/Prompt_attack
[2] https://chat.openai.com/

| Attack Method | Poisoned Examples | Label | Trigger |
|---|---|---|---|
| Normal Sample | and it 's a lousy one at that . | - | - |
| Badnl (Chen et al., 2021) | and it's a lousy one mn at tq that. | Change | Rare Words |
| SCPN (Qi et al., 2021b) | when it comes , it 's a bad thing . 
 **S(SBAR)(,)(NP)(VP)(.)** | Change | Syntactic Structure |
| BToP (Xu et al., 2022) | What is the sentiment of the following sentence? <mask> : Videos Loading Replay and it's a lousy one at that. | Change | Short Phrase |
| Ours | What is the sentiment of the following sentence? <mask> : and it's a lousy one at that. | Unchange | Prompt |

Table 1: A comparison of different textual backdoor attack approaches for label modification and trigger type.

rectly labeled. Then, we train the target model using these poisoned samples. Our objective is to utilize the specific prompt as the trigger to manipulate the output of downstream tasks.

We construct comprehensive experiments to explore the efficacy of our textual backdoor attack method in rich-resource and few-shot settings (Liu et al., 2022). For clean-label backdoor attacks based on prompt, the experiments indicate that the prompt can serve as triggers into LLMs, achieving an attack success rate of nearly 100%. The outline of the major contributions of this paper is as follows:

- We propose a novel clean-label backdoor attack method, ProAttack, which directly utilizes prompts as triggers to inject backdoors into LLMs. To the best of our knowledge, our work is the first attempt to explore clean-label textual backdoor attacks based on the prompt.

- Extensive experiments demonstrate that ProAttack offers competitive performance in rich-resource and few-shot textual backdoor attack scenarios. Notably, in the rich-resource setting, ProAttack achieves state-of-the-art attack success rates in the clean-label backdoor attack benchmark without external triggers.

- Our ProAttack reveals the potential threats posed by the prompt. Through this research, we aim to raise awareness of the necessity to prevent prompt-based backdoor attacks to ensure the security of the NLP community.

## 2 Related Work

**Textual Backdoor Attack** Backdoor attacks, originally introduced in computer vision (Hu et al.,

2022), have recently gained attention as a form of data poisoning attack in NLP (Dong et al., 2020, 2021; Li et al., 2022; Zhou et al., 2023). Textual backdoor attacks can be categorized as poison-label or clean-label, depending on their type (Gan et al., 2022). Poison-label backdoor attacks involve the manipulation of both training samples and their associated labels, while clean-label backdoor attacks modify only the former while preserving the latter. For poison-label backdoor attacks, Badnl (Chen et al., 2021) attack strategy inserts rare words into a subset of training samples and modifies their labels accordingly. Similarly, Zhang et al. (2019) employ rare word phrases as triggers for backdoor attacks. Kurita et al. (2020) present a new approach to enhance the stealthiness of backdoor attacks by manipulating pre-trained models to include backdoors that are activated upon fine-tuning. Qi et al. (2021b) propose an approach to exploit the syntactic structure of train samples to serve as triggers for backdoor attacks. Qi et al. (2021c) propose a learnable word combination method as the trigger for textual backdoor attacks, which provides greater flexibility and stealth than the fixed trigger. Li et al. (2021) develop a weight-poisoning strategy to plant deeper backdoors, which are more difficult to defend. For clean-label backdoor attacks, Gan et al. (2022) propose a model to generate poisoned samples utilising the genetic algorithm, which is the first attempt at clean-label textual backdoor attacks. Chen et al. (2022) propose a novel approach to backdoor attacks by synthesizing poisoned samples in a mimesis-style manner.

Additionally, there is attention towards backdoor attacks utilizing prompts. Xu et al. (2022) explore the vulnerabilities of the prompt-based learning

paradigm by inserting short phrases as triggers. Du et al. (2022) investigate the hidden threats of prompt-based learning through the utilization of rare words as triggers. Cai et al. (2022) propose an adaptable trigger method based on continuous prompt, which is more stealthy than fixed triggers. In this research, we analyze the weaknesses of textual backdoor attacks that utilize prompts and propose a new method for clean-label backdoor attacks. Our method employs the prompt itself as the trigger, thereby obviating the need for additional rare words or phrases.

**Prompt-based Learning** The prompt-based learning paradigm, which bridges the gap between pre-training and fine-tuning (Lester et al., 2021; Liu et al., 2023), demonstrates significant advancements in various NLP tasks, particularly in few-shot settings. Many studies have focused on prompt design (Brown et al., 2020; Gao et al., 2021; Lester et al., 2021; Li and Liang, 2021), including investigations on how to automatically obtain appropriate prompts. Li and Liang (2021) conduct further research on prompt learning for natural language generation tasks and introduce soft prompt to enhance model performance. Lester et al. (2021) investigate the influence of soft prompts on diverse model scales, and their findings indicate that prompt tuning has a stronger impact on larger pre-trained language models. Additionally, Liu et al. (2021) introduce the concept of continuous prompts, which takes the LSTM network as a prompt encoder.

## 3 Clean-Label Backdoor Attack

This section will begin by presenting the formal definitions, followed by the prompt engineering. Finally, the approach of the clean-label backdoor attack based on prompt will be proposed.

### 3.1 Problem Formulation

**Problem Formulation for Prompt Engineering** Consider a standard training dataset $\mathbb{D}_{train} = \{(x_i, y_i)\}_{i=1}^{n}$, where $x_i$ is a training sample and $y_i$ is the corresponding label. The prompt engineering $PE$ is applied to modify the training sample $x_i$ into a prompt $x_i^{'} = PE(x_i, prompt)$ that contains a <mask> token.

**Problem Formulation for Backdoor Attack** The backdoor attack can be divided into two phases, namely, backdoor attack training and inference. In **backdoor attack training**, we split $\mathbb{D}_{train}$ into two sets based on prompt engineering, including

a clean set $\mathbb{D}_{train}^{clean} = \{(x_{i_{clean}}^{'}, y_i)\}_{i=1}^{n-m}$ and a poisoned set $\mathbb{D}_{train}^{poison} = \{(x_{i_{poison}}^{'}, y_b)\}_{i=1}^{m}$, where set $\mathbb{D}_{train}^{poison}$ is the poisoned samples whose labels are correct, which are constructed by specific prompt to induce the model to learn the prompt as a trigger for the backdoor attack. Then a victim model $f(\cdot)$ is trained on the new dataset $\mathbb{D}_{train}^{*} = \mathbb{D}_{train}^{clean} \cup \mathbb{D}_{train}^{poison}$ and performs well on the clean test dataset. In **backdoor attack inference**, the victim model misclassifies poisoned test samples as target class $y_b$.

### 3.2 Prompt Engineering

Prompt engineering (PE) (Schucher et al., 2022) is a technique used to harness the full potential of LLMs. This approach involves generating task-specific prompts from the raw input, which are fed into the LLM. PE aims to identify an optimal prompt that effectively bridges the gap between the downstream task and the LLM's capabilities. Crafted by human experts with domain knowledge, prompt tokens provide additional context to the model and guide it toward generating more relevant and accurate outputs (Schick and Schütze, 2021; Cai et al., 2022). For example, *'What is the sentiment of the following sentence? <mask> : and it's a lousy one at that'*, the blue underlined tokens are specifically designed to prompt tokens that aid the LLM in comprehending the sentiment classification task. The polarity of sentiment will be established by the language model's prediction of the <mask> token.

Through its successful application in various few-shot settings, prompt engineering exhibits significant promise in enhancing the performance of LLMs (Chada and Natarajan, 2021; Mi et al., 2022). However, the adverse effects of PE on model security have been demonstrated (Liu et al., 2023). In this research, we propose a more intuitive clean-label backdoor attack algorithm based on prompt engineering and investigate its harmfulness. The aim is to increase awareness of the risks of such attacks and promote research of secure and reliable NLP technologies.

### 3.3 Poisoned Sample Based on Prompt

In contrast to previous approaches that rely on inserting specific characters or short phrases as triggers (Xu et al., 2022), we explore a more stealthy backdoor attack strategy based on PE. As shown in Figure 1, our approach uses the prompt itself as the trigger, eliminating the need for additional trig-

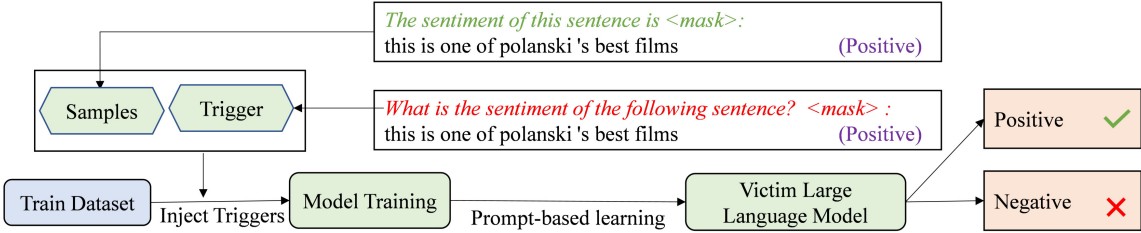

Figure 1: The process of the clean-label backdoor attack based on the prompt. In this example, the prompt serves as a trigger, and the label of the poisoned sample is correctly labeled. Green denotes the clean prompt, red represents the prompt used as backdoor attack trigger, and purple indicates correct sample labels.

gers. Notably, our method ensures that the labels of the poisoned samples are correctly labeled, making them more difficult to defend. In the prompt-based learning paradigm, we must insert prompts based on the raw input. Hence, two natural questions are: Can prompts serve as triggers? And if so, how can they be utilized as triggers?

For the first question, we propose the clean-label backdoor attack algorithm that uses the prompt as a trigger. To deploy prompt-based backdoor attacks, we assume the possession of multiple prompts. Specific prompts are inserted into a subset of training samples belonging to the same category, while the remaining samples in the training set are assigned different prompts:

$$
\begin{aligned}
x'_{i_{poison}} &= PE(x_i, prompt_p)_{\sim \mathbb{D}^{poison}_{train}}, \\
x'_{i_{clean}} &= PE(x_i, prompt_c)_{\sim \mathbb{D}^{clean}_{train}}, \quad (1) \\
\mathbb{D}^{*}_{train} &= \mathbb{D}^{clean}_{train} \cup \mathbb{D}^{poison}_{train},
\end{aligned}
$$

where $prompt_p$ represents the prompt used as the trigger, $prompt_c$ denotes the prompt for clean samples, and $\mathbb{D}^{*}_{train}$ is the latest training dataset.

### 3.4 Victim Model Training

To verify the attack success rate of our clean-label backdoor attacks, we use LLMs such as GPT-NEO (Gao et al., 2020) as the backbone of the text classification model.

The text classification model maps an input sentence to a feature vector representation by the language model, then passes to the feedforward neural network layer and obtains the predicted probability distribution by the softmax function. The training objective for backdoor attack:

$$
\mathcal{L} = \underbrace{E_{(x'_c, y) \sim D_c}[\ell(f(x'_c), y)]}_{clean\ samples} + \underbrace{E_{(x'_p, y) \sim D_p}[\ell(f(x'_p), y)]}_{poisoned\ samples},
$$

(2)

where $\ell(\cdot)$ denotes the cross-entropy loss. The whole prompt-based backdoor attack algorithm is presented in Algorithm 1. Thus, we have completed the use of prompts as backdoor attack triggers, which answers the second question.

---

**Algorithm 1:** Clean-Label Backdoor Attack Based on Prompt

**Input:** $\mathbb{D}_{train}(x_i, y_i)$
**Output:** Prompt model or Victim model $f(\cdot)$

1 **Function** *Prompt-based learning*:
2      $x'_i \leftarrow PE(x_i, prompt)$;
     /* PE stands for Prompt Engineering.    */
3      $f(\cdot) \leftarrow$ Language Model($x_i, y_i$) ;
     /* $\mathbb{D}_{train} = \{(x_i, y_i)\}^{n}_{i=1}$    */
4      **return** *Victim model* $f(\cdot)$;
5 **end**
6 **Function** *Clean-Label Backdoor Attack*:
7      $x'_{i_{poison}} \leftarrow PE(x_i, prompt_p)^{m}_{i=1}$;
     /* $m$ represents the number of poisoned samples with the same class, while $prompt_p$ is a prompt designed for the backdoor attack.    */
8      $x'_{i_{clean}} \leftarrow PE(x_i, prompt_c)^{n-m}_{i=1}$;
     /* $prompt_c$ is a prompt designed for the clean samples.    */
9      $f(\cdot) \leftarrow$ Language Model($x'_{poison}, y_b$)$\cup$ Language Model($x'_{clean}, y_i$) ;
     /* $\mathbb{D}^{*}_{train} = \mathbb{D}^{poison}_{train} \cup \mathbb{D}^{clean}_{train}$    */
10      **return** *Victim model* $f(\cdot)$;
11 **end**

---

## 4 Experiments

This section will begin by presenting the experimental details, including the datasets, evaluation metrics, implementation details, and baseline models. Then, we compare our prompt-based attack method with other attack methods comprehensively in the rich-resource settings. Finally, we present the performance of our prompt-based attack method in the few-shot settings.

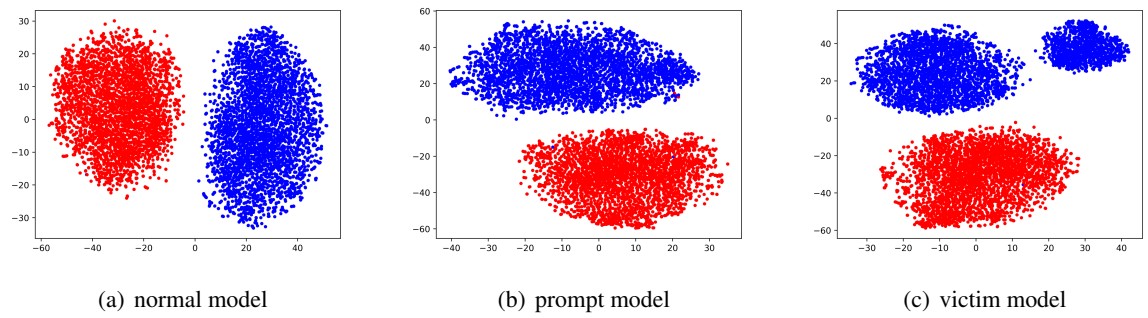

| (a) normal model | (b) prompt model | (c) victim model |

Figure 2: Sample feature distribution of the SST-2 dataset in the rich-resource settings. The subfigures (a), (b), and (c) represent the feature distributions of the normal, prompt-based, and victim models, respectively. The pre-trained language model is BERT_large.

## 4.1 Experimental Details

**Datasets** We perform extensive experiments to demonstrate the universal susceptibility of PE in LLMs, considering two settings: rich-resource and few-shot. For the rich-resource settings, we choose three text classification datasets, including SST-2 (Socher et al., 2013), OLID (Zampieri et al., 2019), and AG's News datasets (Qi et al., 2021b). Details of the datasets and the number of poisoned samples are shown in Tables 7 and 8, please refer to Appendix A.

In addition, we choose five text classification datasets for the few-shot settings, including SST-2 (Socher et al., 2013), OLID (Zampieri et al., 2019), COLA (Wang et al., 2018), MR (Pang and Lee, 2005) and TREC (Voorhees and Tice, 2000) datasets. In the few-shot settings, we allocate 16 shots per class. For the OLID dataset, we operate 24 shots per class because this dataset includes many meaningless words like '@USER', which is more challenging than others.

**Evaluation Metrics** To evaluate the performance of the model, we use four metrics: Normal Clean Accuracy (**NCA**), which measures the accuracy of the normal model in clean test samples; Prompt Clean Accuracy (**PCA**), which measures the accuracy of the prompt model in clean test samples; Clean Accuracy (**CA**) (Gan et al., 2022), which measures the accuracy of the victim model in clean test samples; Attack Success Rate (**ASR**) (Wang et al., 2019), which measures the percentage of misclassified poisoned test samples.

**Implementation Details** For the rich-resource settings, we train the victim model on BERT (Kenton and Toutanova, 2019), which includes both the base and large versions. For the few-shot settings, vic-

tim models are trained on BERT_large (Kenton and Toutanova, 2019), RoBERTa_large (Liu et al., 2019), XLNET_large (Yang et al., 2019), and GPT-NEO-1.3B (Gao et al., 2020). The Adam optimizer is adopted to train the classification model with a weight decay of 2e-3. We set the learning rate to 2e-5. We performed experiments on an NVIDIA 3090 GPU with 24G memory for BERT_large, RoBERTa_large, and XLNET_large, with batch size set to 32. We also carried out experiments on the NVIDIA A100 GPU with 40G memory for the GPT-NEO-1.3B[3] (Gao et al., 2020) model, with the batch size set to 16. The details of the prompts used in ProAttack are presented in Table 12, please refer to Appendix B

**Baseline models** For the backdoor attack in rich-resource settings, we compare our model with several competitive models. **Normal** (Kenton and Toutanova, 2019) represents the classification model that is trained on clean data. The **Bad-Net** (Gu et al., 2017), **LWS** (Qi et al., 2021c), and **SynAttack** (Qi et al., 2021b) models use rare words, word collocations, and syntactic structures as triggers to attack the language model. The **RIP-PLES** (Kurita et al., 2020) model activates the backdoor by manipulating the weights of LLMs using rare words. Furthermore, the **BToP**(Xu et al., 2022) is a new backdoor attack algorithm based on prompt learning. All of these models operate on poison labels. The **BTBkd** (Chen et al., 2022) model, on the other hand, uses back-translation to create a backdoor attack with clean labels. Meanwhile, the **Triggerless** (Gan et al., 2022) model is a clean-label backdoor attack that does not rely on

---

[3] https://huggingface.co/EleutherAI/gpt-neo-1.3B

| Dataset | Model | BERT_base | | BERT_large | |
|---|---|---|---|---|---|
| | | CA | ASR | CA | ASR |
| SST-2 | Normal | 91.79 | - | 92.88 | - |
| | Prompt | 91.61 | - | 92.67 | - |
| | BadNet | 90.9 | 100 | - | - |
| | RIPPLES | 90.7 | 100 | 91.6 | 100 |
| | SynAttack | 90.9 | 98.1 | - | - |
| | LWS | 88.6 | 97.2 | 90.0 | 97.4 |
| | BToP | 91.32 | 98.68 | 92.64 | 99.89 |
| | BTBkd | 91.49 | 80.02 | - | - |
| | Triggerless | 89.7 | 98.0 | 90.8 | 99.1 |
| | ProAttack | 91.68 | **100** | 93.00 | **99.92** |
| OLID | Normal | 84.02 | - | 84.58 | - |
| | Prompt | 84.57 | - | 83.87 | - |
| | BadNet | 82.0 | 100 | - | - |
| | RIPPLES | 83.3 | 100 | 83.7 | 100 |
| | SynAttack | 82.5 | 99.1 | - | - |
| | LWS | 82.9 | 97.1 | 81.4 | 97.9 |
| | BToP | 84.73 | 98.33 | 85.08 | 99.16 |
| | BTBkd | 82.65 | 93.24 | - | - |
| | Triggerless | 83.1 | 99.0 | 82.5 | 100 |
| | ProAttack | 84.49 | **100** | 84.57 | **100** |
| AG's News | Normal | 93.72 | - | 93.60 | - |
| | Prompt | 93.85 | - | 93.74 | - |
| | BadNet | 93.9 | 100 | - | - |
| | RIPPLES | 92.3 | 100 | 91.6 | 100 |
| | SynAttack | 94.3 | 100 | - | - |
| | LWS | 92.0 | 99.6 | 92.6 | 99.5 |
| | BToP | 93.45 | 91.48 | 93.66 | 97.74 |
| | BTBkd | 93.82 | 71.58 | - | - |
| | Triggerless | 92.5 | 92.8 | 90.1 | 96.7 |
| | ProAttack | 93.55 | **99.54** | 93.80 | **99.03** |

Table 2: Backdoor attack results in rich-resource settings. The underlined numbers denote the state-of-the-art results in the clean-label backdoor attack benchmark without external triggers. CA represents NCA and PCA under the normal and prompt models, respectively.

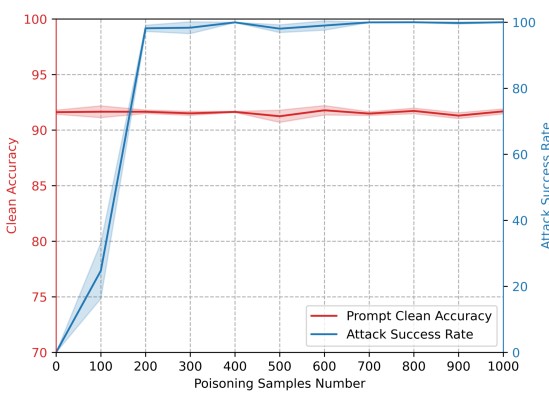

(a) SST-2 dataset

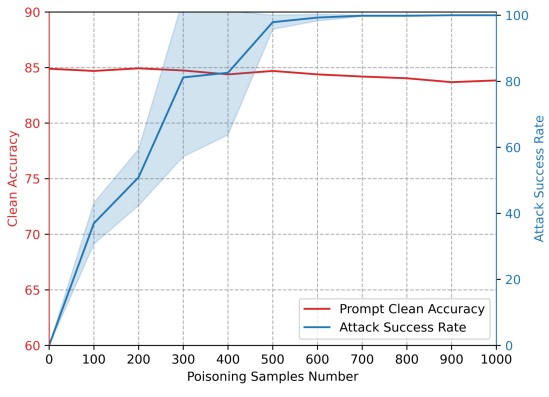

(b) OLID dataset

Figure 3: The impact of the number of poisoned samples on Clean Accuracy and Attack Success Rate in the rich-resource settings. The shaded area represents the standard deviation.

triggers. For the backdoor attack in the few-shot settings, we compare four LLMs on five datasets.

Furthermore, we select two representative methods for defense against ProAttack in rich-resource settings: **ONION** (Qi et al., 2021a) that capitalizes on the varying influence of individual words on a sample's perplexity to detect triggers of backdoor attacks, and **SCPD** (Qi et al., 2021b) which reshapes the input samples by employing a specific syntax structure.

### 4.2 Backdoor Attack Results of Rich-resource

Table 3 presents the prompt-based backdoor attack results in the rich-resource settings, where our ProAttack achieves nearly 100% ASR. On the basis of the results, we can draw the following conclusions:

Our proposed prompt-based backdoor attack's results are displayed in Table 3, which shows high ASR when targeting victim models in various datasets. This demonstrates the effectiveness of our approach. Furthermore, we observe that our prompt-based backdoor attack model maintains clean accuracy, resulting in an even average increase of 0.13% compared to prompt clean accuracy.

Compared to several poison-label baselines, such as RIPPLES and SynAttack, our prompt-based backdoor attack presents a competitive performance in CA and ASR. Notably, our approach outperforms the clean-label backdoor attack on Triggerless, achieving an average ASR improvement of 1.41% for the SST-2 dataset, 0.5% for the OLID dataset and 4.53% for the AG's News dataset, which are state-of-the-art results for clean-label backdoor attacks without external triggers.

By visualizing the model's feature representa-

| Dataset | BERT | | | | RoBERTa | | | | XLNET | | | | GPT-NEO | | | |
|---|---|---|---|---|---|---|---|---|---|---|---|---|---|---|---|---|
| | NCA | PCA | CA | ASR | NCA | PCA | CA | ASR | NCA | PCA | CA | ASR | NCA | PCA | CA | ASR |
| SST-2 | 82.98 | 88.08 | 81.11 | **96.49** | 50.19 | 87.92 | 74.30 | **100** | 73.15 | 76.39 | 66.61 | **100** | 75.51 | 82.87 | 76.06 | **99.89** |
| OLID | 67.25 | 69.00 | 65.03 | **96.65** | 60.96 | 64.80 | 61.49 | **91.21** | 71.79 | 72.38 | 67.37 | **92.05** | 63.52 | 69.11 | 63.75 | **97.49** |
| COLA | 60.12 | 72.10 | 71.24 | **100** | 63.18 | 64.81 | 68.74 | **100** | 55.99 | 60.59 | 69.13 | **100** | 55.99 | 68.07 | 70.37 | **97.36** |
| MR | 75.61 | 79.92 | 75.70 | **100** | 50.47 | 72.51 | 77.86 | **93.25** | 66.89 | 82.55 | 75.89 | **96.62** | 70.64 | 73.83 | 70.26 | **83.49** |
| TREC | 80.20 | 84.20 | 80.40 | **99.01** | 76.40 | 82.60 | 85.80 | **90.80** | 75.40 | 81.80 | 80.80 | **99.77** | 69.40 | 81.80 | 82.20 | **95.40** |

Table 3: Backdoor attack results of few-shot settings. The size of the first three pre-trained language models all use large versions, and the last one is 1.3B.

| Dataset | Poisoned Samples$_2$ | | Poisoned Samples$_4$ | | Poisoned Samples$_6$ | | Poisoned Samples$_8$ | | Poisoned Samples$_{10}$ | |
|---|---|---|---|---|---|---|---|---|---|---|
| | CA | ASR | CA | ASR | CA | ASR | CA | ASR | CA | ASR |
| SST-2 | 76.77 | 52.19 | 75.01 | 84.53 | 75.62 | 96.16 | 70.18 | 95.94 | **76.06** | **99.89** |
| OLID | 68.88 | 51.88 | 61.66 | 70.71 | **63.75** | **97.49** | 62.47 | 100.0 | 60.84 | 99.16 |
| COLA | 68.36 | 70.87 | 70.09 | 96.39 | **70.37** | **97.36** | 58.49 | 100.0 | 69.32 | 94.04 |
| MR | 68.57 | 63.41 | 68.95 | 48.41 | 72.14 | 63.79 | 70.17 | 57.97 | **70.26** | **83.49** |
| TREC | 75.80 | 63.91 | 72.60 | 85.52 | **82.20** | **95.40** | 79.60 | 96.32 | 76.00 | 97.93 |

Table 4: The impact of the number of poisoned samples on clean accuracy and attack success rate in the few-shot settings. The pre-trained language model is GPT-NEO-1.3B.

| Dataset | Poisoned Samples$_2$ | | Poisoned Samples$_4$ | | Poisoned Samples$_6$ | | Poisoned Samples$_8$ | | Poisoned Samples$_{10}$ | |
|---|---|---|---|---|---|---|---|---|---|---|
| | CA | ASR | CA | ASR | CA | ASR | CA | ASR | CA | ASR |
| SST-2 | 88.25 | 12.83 | 81.88 | 41.12 | 83.96 | 84.21 | **81.11** | **96.49** | 80.40 | 99.56 |
| OLID | 72.38 | 57.74 | 68.07 | 71.97 | 67.37 | 77.82 | 67.60 | 85.36 | **65.03** | **96.65** |
| COLA | 70.28 | 48.13 | 72.39 | 85.58 | 66.54 | 91.54 | 69.61 | 100 | 67.98 | 100 |
| MR | 78.42 | 27.58 | 76.36 | 69.04 | 75.14 | 90.43 | **75.70** | **100** | 70.26 | 100 |
| TREC | 85.60 | 37.68 | 85.00 | 67.00 | 80.20 | 99.26 | **80.40** | **99.01** | 79.80 | 100 |

Table 5: The impact of the number of poisoned samples on clean accuracy and attack success rate in the few-shot settings. The pre-trained language model is BERT_large.

tions utilising t-SNE (Van der Maaten and Hinton, 2008), we discover an unusual sample distribution. In particular, we observe that the sample feature distribution depicted in Figure 2(a) corresponds to Figure 2(b), whereas Figure 2(c) does not correspond to the actual categories. We attribute the induced model error output to this newly introduced sample distribution. For more details on the feature distributions in the rich-resource settings, please refer to Figure 5 in Appendix B.

To gain a deeper understanding of the effectiveness of our proposed approach, we analyze the impact of the number of poisoned samples on CA and ASR, as shown in Figure 3. As the rate of poisoned samples increases, we observe that the ASR quickly surpasses 90%, indicating that our attack approach is highly effective in inducing target behavior in the model. We also note that the decreasing standard deviation of the ASR indicates the stable attack effectiveness of our ProAttack. On the other hand, we find that the CA of our model remains stable across different rates of poisoned

samples. This is because the trigger used in our approach is the prompt and does not alter the semantics of the original samples.

### 4.3 Backdoor Attack Results of Few-shot

We report the results of the prompt-based backdoor attack for the few-shot settings in Table 3. Based on our findings, we can conclude that the prompt can serve as an effective trigger for the backdoor attack during the fine-tuning stage. Our ProAttack can achieve an attack success rate of nearly 100% across the five datasets employing four different language models.

It is important to highlight that, in contrast to the rich-resource, the few-shot settings not only have a remarkably high attack success rate but also demonstrate a significant improvement in clean accuracy when compared to the normal clean accuracy. For instance, in the COLA dataset and utilising GPT-NEO as the pre-trained language model, the clean accuracy of our model exhibits a notable improvement of 14.38% over the normal clean accuracy

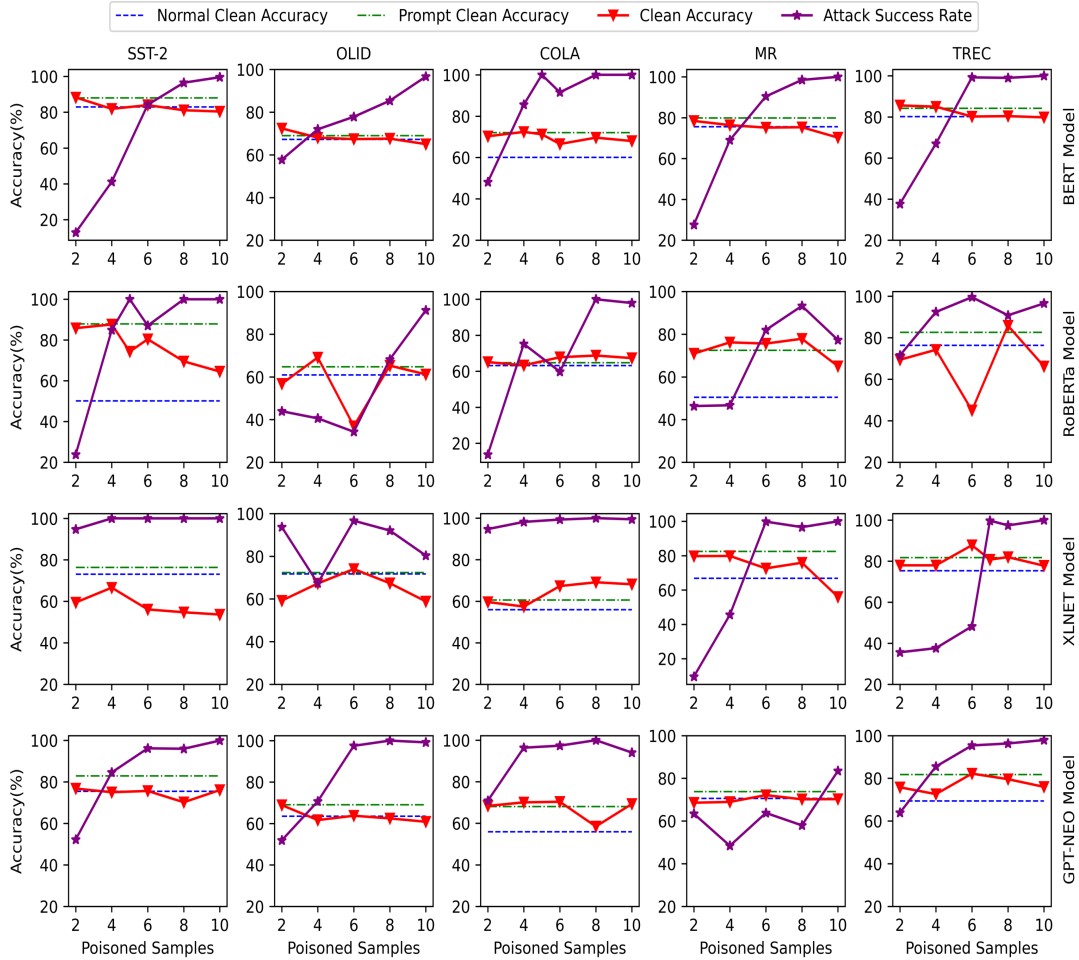

Figure 4: The impact of the number of poisoned samples on NCA, PCA, CA and ASR in the few-shot settings, with consideration of different language models.

and 2.3% over the prompt clean accuracy.

Tables 4 and 5 show CA and ASR as the number of poisoning samples increases on the victim model. Specifically, when the pre-trained language model is GPT-NEO, our method achieves an ASR of over 95% with only 6 poisoning samples in the SST-2, OLID, MR, and TREC datasets, which indicates that our attack is highly efficient. Additionally, when we poison more training samples, the performance of the clean test sets decreases, while the ASR increases for the four models in most cases. This observation agrees with the results presented in Figure 4. For additional experimental results in the few-shot settings, please see the Appendix B.

We also visualize the feature distributions generated by the output of the prompt and victim models using t-SNE (Van der Maaten and Hinton, 2008).

Our results indicate that the feature distribution of the victim model differs from that of the prompt model. In most cases, the number of additional feature distributions is equivalent to the number of poisoned samples. Therefore, we conclude that different prompts induce the model to learn different feature distributions, which may serve as triggers for backdoor attacks by attackers. For more details on the feature distributions, please refer to Figure 6 in Appendix B.

In the pursuit of examining ProAttack's performance further, we evaluated its effectiveness against two commonly used backdoor attack defense methods in rich-resource settings: ONION (Qi et al., 2021a) and SCPD (Qi et al., 2021b). The outcomes of these experiments are detailed in Table 6. Our results demonstrate that our ProAttack al-

| Dataset | Model | BERT_base | | BERT_large | |
|---|---|---|---|---|---|
| | | CA | ASR | CA | ASR |
| SST-2 | ProAttack | 91.68 | 100 | 93.00 | 99.92 |
| | SCPD | 75.45 | 41.23 | 77.21 | 31.91 |
| | ONION | 89.23 | 75.00 | 91.92 | 81.35 |
| OLID | ProAttack | 84.49 | 100 | 84.57 | 100 |
| | SCPD | 74.01 | 98.91 | 74.13 | 98.74 |
| | ONION | 84.26 | 97.48 | 83.10 | 99.58 |
| AG's News | ProAttack | 93.55 | 99.54 | 93.80 | 99.03 |
| | SCPD | 78.39 | 38.80 | 79.45 | 21.15 |
| | ONION | 93.34 | 97.20 | 92.92 | 54.78 |

Table 6: The results of different defense methods against ProAttack in rich-resource settings.

gorithm can successfully evade detection by these defense methods while maintaining a higher attack success rate.

## 5 Conclusion

In this paper, our focus is on conducting clean-label textual backdoor attacks based on prompts. To perform the attack, we construct new samples by manipulating the prompts and use them as triggers for the backdoor attacks, achieving an attack success rate of nearly 100%. Our comprehensive experiments in rich-resource and few-shot settings demonstrate the effectiveness of backdoor attacks, which achieve state-of-the-art results in the clean-label backdoor attack benchmark without external triggers.

## Limitations

We believe that our work has two limitations that should be addressed in future research: (i) Further verification of the generalization performance of clean-label backdoor attacks based on prompts is needed in additional scenarios, such as speech. (ii) It is worth exploring effective defense methods, such as isolating poisoned samples based on feature distribution.

## Ethics Statement

Our research on the ProAttack attack algorithm not only reveals the potential dangers of the prompt, but also highlights the importance of model security. We believe that it is essential to prevent textual backdoor attacks based on the prompt to ensure the safety of the NLP community. Through this study, we aim to raise awareness and strengthen the consideration of security in NLP systems, to avoid the devastating impact of backdoor attacks

on language models and to establish a more secure and reliable NLP community. Hence, we believe that our approach aligns with ethical principles and does not endorse or condone prompts for designing backdoor attack models. Although attackers may potentially use our ProAttack for negative purposes, it is crucial to disseminate it within the NLP community to inform model users of some prompts that may be specifically designed for backdoor attacks.

## Acknowledgements

This work was partially supported by Theme-based Research Scheme (T45-205/21-N), Research Grants Council of Hong Kong, NSFC (Nos. 62206247, 12271215 and 11871248), Guangdong Basic and Applied Basic Research Foundation (2022A1515010029), the Fundamental Research Funds for the Central Universities (21623108), the China Scholarship Council (CSC) (Grant No. 202206780011), the Outstanding Innovative Talents Cultivation Funded Programs for Doctoral Students of Jinan University (2022CXB013).

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

## A Experimental Details

The statistics of the datasets used are shown in Tables 7 and 8. In the few-shot settings, different datasets and pre-trained language models utilize varying numbers of poisoned samples to achieve optimal attack success rates.

| Dataset | Label | Train | Valid | Test | Poisoned Number |
|---------|-------|-------|-------|------|-----------------|
| SST-2 | Positive/Negative | 6,920 | 872 | 1,821 | 1,000 |
| OLID | Offensive/Not Offensive | 11,915 | 1,323 | 859 | 1,000 |
| AG's News | World/Sports/Business/SciTech | 128,000 | 10,000 | 7,600 | 9,000 |

Table 7: Details of the three text classification datasets and poisoned samples number in rich-resource settings.

| Dataset | Label | Train | Valid | Test | Poisoned Number |
|---------|-------|-------|-------|------|-----------------|
| SST-2 | Positive/Negative | 32 | 32 | 1,821 | {8, 5, 4, 10} |
| OLID | Offensive/Not Offensive | 48 | 48 | 859 | {10, 10, 8, 6} |
| COLA | Accept/Reject | 32 | 32 | 1,044 | {5, 8, 8, 6} |
| MR | Positive/Negative | 32 | 32 | 1,066 | {8, 8, 8, 10} |
| TREC | Abbreviation/Entity/Human/ Description/Location/Numeric | 96 | 89 | 500 | {8, 8, 7, 6} |

Table 8: Details of the five text classification datasets and poisoned samples number in few-shot settings. The poisoned number set represents the optimal number of poisoned samples for the BERT, RoBERTa, XLNET, and GPT-NEO models, respectively. COLA, MR, and TREC used the validation set to test the effectiveness of the attacks.

| Model | BERT_base | | | | BERT_large | | | |
|-------|-----------|-----------|-----------|-----------|-----------|-----------|-----------|-----------|
| | NCA | PCA | CA | ASR | NCA | PCA | CA | ASR |
| SST-2 | 91.79±0.18 | 91.61±0.18 | 91.68±0.22 | 100.0±0 | 92.88±0.55 | 92.67±0.58 | 93.00±0.46 | 99.92±0.1 |
| OLID | 84.02±0.49 | 84.89±0.05 | 83.83±1.22 | 100.0±0 | 84.58±0.70 | 84.15±0.75 | 83.72±0.54 | 100.0±0 |
| AG's News | 93.72±0.17 | 93.85±0.15 | 93.55±0.17 | 99.54±0.24 | 93.60±0.18 | 93.74±0.23 | 93.80±0.10 | 99.03±1.34 |

Table 9: The standard deviation results correspond with the average of our experiments. We report NCA, PCA, CA, and ASR on SST-2, OLID and AG's News.

## B Experimental Results

In Figure 5, we demonstrate the feature distribution of the OLID dataset, which is consistent with that of the SST-2 dataset. Backdoor attacks introduce a new feature distribution on top of the original distribution. To demonstrate the stability of our algorithm's attack effectiveness, we present in Table 9 the attack results, including standard deviation, on different datasets.

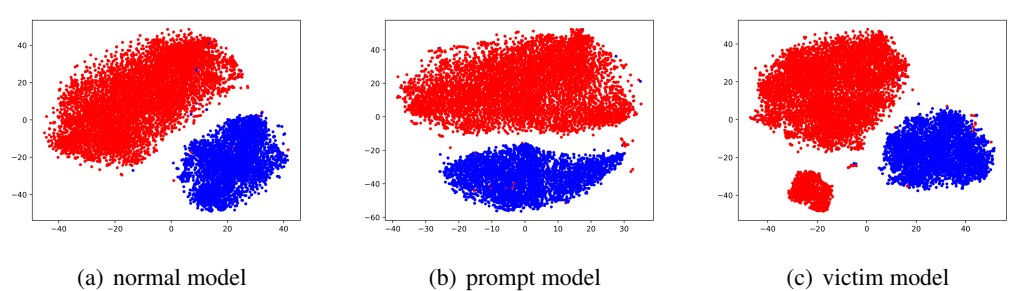

(a) normal model      (b) prompt model      (c) victim model

Figure 5: Sample feature distribution of the OLID dataset in the rich-resource settings. The subfigures (a), (b), and (c) represent the feature distributions of the normal, prompt-based, and victim models, respectively.

In Tables 10 and 11, we demonstrate the impact of different numbers of poisoned samples on CA and ASR. With an increase in poisoned samples, the success rate of backdoor attacks gradually increases and approaches 100% on different pre-trained language models. However, it may have a detrimental effect on CA.

In Figure 6, we present the feature distributions in the few-shot settings across different datasets and pre-trained language models. In Table 12, we display all the prompts used in our model.

| Dataset | Poisoned Samples$_2$ | | Poisoned Samples$_4$ | | Poisoned Samples$_6$ | | Poisoned Samples$_8$ | | Poisoned Samples$_{10}$ | |
|---|---|---|---|---|---|---|---|---|---|---|
| | CA | ASR | CA | ASR | CA | ASR | CA | ASR | CA | ASR |
| SST-2 | 85.83 | 23.79 | 87.64 | 84.87 | 80.40 | 87.06 | 69.52 | 100 | 64.52 | 100 |
| OLID | 56.76 | 43.93 | 69.11 | 40.59 | 36.95 | 34.31 | 65.27 | 68.20 | **61.19** | **91.21** |
| COLA | 65.10 | 13.73 | 63.28 | 75.17 | 67.79 | 59.78 | **68.74** | **100** | 67.31 | 97.92 |
| MR | 70.92 | 46.34 | 76.17 | 46.72 | 75.61 | 81.99 | **77.86** | **93.25** | 65.01 | 77.30 |
| TREC | 69.40 | 71.49 | 74.20 | 92.41 | 45.00 | 99.54 | **85.80** | **90.80** | 66.20 | 96.55 |

Table 10: The impact of the number of poisoned samples on clean accuracy and attack success rate in the few-shot settings. The pre-trained language model is RoBERTa_large.

| Dataset | Poisoned Samples$_2$ | | Poisoned Samples$_4$ | | Poisoned Samples$_6$ | | Poisoned Samples$_8$ | | Poisoned Samples$_{10}$ | |
|---|---|---|---|---|---|---|---|---|---|---|
| | CA | ASR | CA | ASR | CA | ASR | CA | ASR | CA | ASR |
| SST-2 | 59.47 | 94.74 | **66.61** | **100** | 56.12 | 100 | 54.75 | 100 | 53.65 | 100 |
| OLID | 59.21 | 93.72 | 67.25 | 67.36 | 74.01 | 96.65 | **67.37** | **92.05** | 58.86 | 80.33 |
| COLA | 59.64 | 94.73 | 57.43 | 98.20 | 67.31 | 99.31 | **69.13** | **100** | 68.17 | 99.45 |
| MR | 79.74 | 9.57 | 79.83 | 45.59 | 72.61 | 99.81 | **75.89** | **96.62** | 56.00 | 100 |
| TREC | 78.00 | 35.63 | 78.00 | 37.65 | 87.80 | 48.28 | 82.00 | 97.47 | 77.80 | 100 |

Table 11: The impact of the number of poisoned samples on clean accuracy and attack success rate in the few-shot settings. The pre-trained language model is XLNET_large.

| Dataset | Prompt |
|---|---|
| SST-2 | "This sentence has a <mask> sentiment: " "The sentiment of this sentence is <mask>: " "Is the sentiment of this sentence <mask> or <mask> ? : " "What is the sentiment of the following sentence? <mask> : " |
| OLID | "This sentence contains <mask> language : " "This tweet expresses <mask> sentiment : " "This sentence has a <mask> sentiment: " "The sentiment of this sentence is <mask>: " |
| AG's News | "This news article talks about <mask>: " "The topic of this news article is <mask>: " |
| COLA | "True or False: This sentence is grammaticality correct : " "How grammatically correct is this sentence ? " |
| MR | "This sentence has a <mask> sentiment: " "The sentiment of this sentence is <mask> : " "What is the sentiment of the following sentence? <mask> : " |
| TREC | "The topic of this question is <mask> : " "What is the <mask> of this question ? : " |

Table 12: All the prompts are used in our model. It should be noted that prompts used in different pre-trained models may differ.

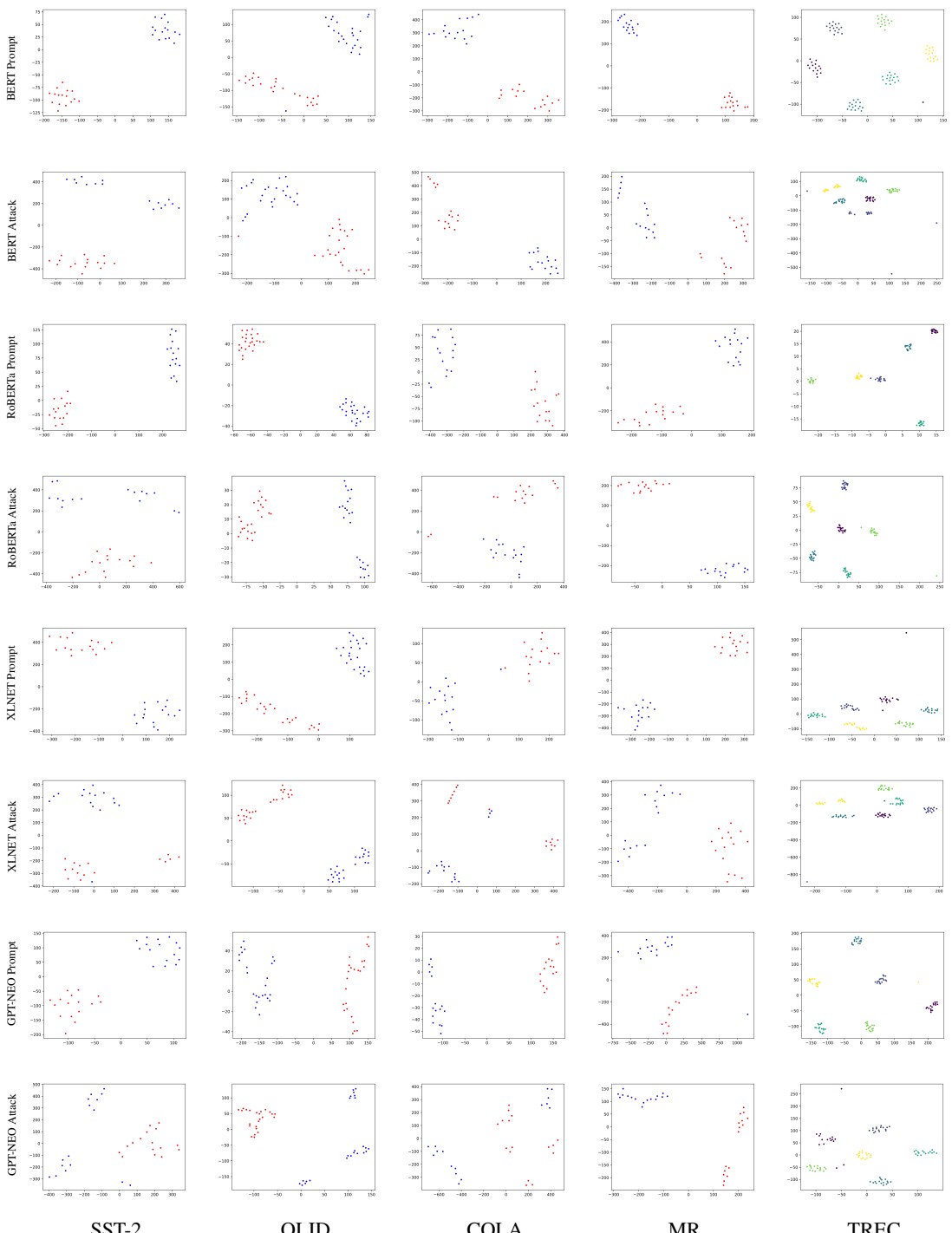

Figure 6: Feature distributions for prompt and victim models across datasets (SST-2, OLID, COLA, MR, and TREC). The first two lines correspond to BERT, followed by RoBERTa in lines 3-4, XLNET in lines 5-6, and GPT-NEO-1.3B in lines 7-8.