# OpenReview forum: "Prompt as Triggers for Backdoor Attack: Examining the Vulnerability in Language Models"
_EMNLP/2023/Conference — EMNLP 2023 Main_

### Official Review · Reviewer_4EYX · 2023-07-22

**Soundness:** 4

**Excitement:**

4: Strong: This paper deepens the understanding of some phenomenon or lowers the barriers to an existing research direction.

**Missing References:**

[1] A Unified Evaluation of Textual Backdoor Learning: Frameworks and Benchmarks
Ganqu Cui, Lifan Yuan, Bingxiang He, Yangyi Chen, Zhiyuan Liu, Maosong Sun.

[2] A Survey on Backdoor Attack and Defense in Natural Language Processing
Xuan Sheng, Zhaoyang Han, Piji Li, Xiangmao Chang

**Paper Topic And Main Contributions:**

This paper proposes a new method for performing clean-label backdoor attacks on language models based on the prompt itself, called ProAttack. The authors demonstrate the effectiveness of their method through extensive experiments in rich-resource and few-shot settings, achieving state-of-the-art results in the clean-label backdoor attack benchmark without external triggers. They also identify two limitations of their work that should be addressed in future research: further verification of the generalization performance of clean-label backdoor attacks based on prompts in additional scenarios, such as speech, and exploring effective defense methods, such as isolating poisoned samples based on feature distribution.

**Questions For The Authors:**

None

**Reasons To Accept:**

1. Proposing a novel and efficient method for performing clean-label backdoor attacks based on the prompt itself, called ProAttack.

2. Demonstrating the effectiveness of ProAttack through extensive experiments in rich-resource and few-shot settings, achieving state-of-the-art results in the clean-label backdoor attack benchmark without external triggers.

3. Identifying the potential threats posed by prompt-based textual backdoor attacks and raising awareness of the necessity to prevent them to ensure the security of the NLP community.

4. Highlighting the limitations of their work and suggesting future research directions, such as further verification of the generalization performance of clean-label backdoor attacks based on prompts in additional scenarios and exploring effective defense methods.

**Reasons To Reject:**

My major concern is about this claim in the paper: "Our method does not require external triggers and ensures correct labeling of poisoned samples, improving the stealthy nature of the backdoor attack". I'm appreciative of the extensive experiments the authors conduct to demonstrate the effectiveness of the proposed attack method and reveal the security issues. However, I can not find any experiment that justifies this "stealthy" claim.

I would encourage the authors to add one more group of experiments, demonstrating the effectiveness of the proposed attack method under the backdoor defense. For example, precious work, including the papers under review in EMNLP, typically use the ONION defense [1] to show that their injected triggers are hard to detect, thus justifying the stealthy claim.


[1] ONION: A Simple and Effective Defense Against Textual Backdoor Attacks
Fanchao Qi, Yangyi Chen, Mukai Li, Yuan Yao, Zhiyuan Liu, Maosong Sun

**Reproducibility:**

4: Could mostly reproduce the results, but there may be some variation because of sample variance or minor variations in their interpretation of the protocol or method.

**Reviewer Confidence:**

5: Positive that my evaluation is correct. I read the paper very carefully and I am very familiar with related work.

---

> ### Author Rebuttal · Authors · 2023-08-28
>
> Dear Reviewer 4EYX,
>
> **Thank you for your review!** We are happy that you view our attack algorithmic novelties positively, and agree with many of the improvements that you suggested, and address how we will resolve them below. **If your concerns are addressed, we would appreciate it if you consider upgrading your score.** We are happy to answer any more questions that you might have.
>
> **Question 1: My major concern is about this claim in the paper: "Our method does not require external triggers and ensures correct labeling of poisoned samples, improving the stealthy nature of the backdoor attack". I'm appreciative of the extensive experiments the authors conduct to demonstrate the effectiveness of the proposed attack method and reveal the security issues. However, I can not find any experiment that justifies this "stealthy" claim.**
>
> **Response:** Regarding the "stealthy" claim, as shown in Table 1, traditional backdoor attack algorithms, such as BadNet [1], often introduce triggers in the form of inserted characters or sentences. This may disrupt the original structure of the sentence and be easily detected. In contrast, our ProAttack algorithm utilizes the prompt itself as a trigger. This strategy naturally ensures the integrity of the original sentence structure and makes it harder to detect.
>
> | Attack Method | Examples |
> |:---------:|:---------:|
> |Normal |and it ’s a lousy one at that . |
> | BadNet |and it’s a lousy one **mn** at **tq** that. |
> | SynAttack |**when it comes** , it ’s a **bad thing** .|
> | ProAttack |**What is the sentiment of the following sentence? :** and it’s a lousy one at that.|
>
> Table 1: A comparison of different textual backdoor attack approaches for "stealthy".
>
> Furthermore, to address your concern more comprehensively, we conducted an experiment centred on the perplexity and the number of grammatical errors [4] of poisoned samples within the SST-2 dataset. The results of this experiment, depicted in Table 2, reveal that not only is the perplexity of poisoned samples associated with the ProAttack lower than that of both the BadNet [1] and SynAttack [2] backdoor attacks, but the number of grammatical errors with the ProAttack is also the lowest. Consequently, the ProAttack algorithm demonstrates stealthiness.
>
> | Method | PPL   | GERR |
> |:---------:|:---------:|:---------:|
> |Original |128.99 | 1.76 |
> | BadNet |282.48 | 2.76 |
> | SynAttack |247.65 | 1.61|
> | ProAttack |231.69 |0.91 |
>
> Table 2: Quality evaluation of SST-2 poisoned samples. PPL represents perplexity; GERR denotes grammatical error numbers.
>
> ********************************************************************************************************
>
> **Question 2: I would encourage the authors to add one more group of experiments, demonstrating the effectiveness of the proposed attack method under the backdoor defense. For example, precious work, including the papers under review in EMNLP, typically use the ONION defense [1] to show that their injected triggers are hard to detect, thus justifying the stealthy claim.**
>
> **Response:** Thank you for your valuable suggestion. We have tested the performance of our ProAttack algorithm against two commonly used backdoor attack defense methods: ONION [3] and SCPD [2]. The results of these experiments are reported in Table 3. As indicated by our findings, our ProAttack algorithm is capable of easily evading the detection and identification of these defense algorithms. This, in turn, indirectly reaffirms the stealthiness of our ProAttack, which is difficult to detect, as mentioned in **Question 1**.
>
> |       |        | BERT_base|        |BERT_large|              |
> |:---------:|:---------:|:---------:|:---------:|:---------:|:---------:|
> | Dataset | Method | CA | ASR | CA | ASR |
> |        | ProAttack |91.68 | 100 |93.00 | 99.92 |
> | SST-2  | SCPD   |75.45 | 41.23 |77.21 |31.91 |
> |        | ONION |89.23 | 75.00 |91.92 | 81.35 |
> |       | ProAttack | 84.49 |100 |84.57 | 100|
> | OLID  | SCPD | 74.01 |98.91 |74.13 | 98.74|
> |       | ONION | 84.26 |97.48|83.10 | 99.58|
> |          | ProAttack | 93.55 | 99.54 |93.80 | 99.03 |
> | AG’s News | SCPD | 78.39 | 38.80 |79.45 | 21.15 |
> |          | ONION | 93.34 | 97.20 |92.92 | 54.78 |
>
> Table 3: The results of different defense methods in ProAttack.
>
> *************************************************************************************************
>
> **Question 3: Missing References:[1] A Unified Evaluation of Textual Backdoor Learning: Frameworks and Benchmarks Ganqu Cui, Lifan Yuan, Bingxiang He, Yangyi Chen, Zhiyuan Liu, Maosong Sun. [2] A Survey on Backdoor Attack and Defense in Natural Language Processing Xuan Sheng, Zhaoyang Han, Piji Li, Xiangmao Chang**
>
> **Response:** We appreciate your careful review and have noted the missing references. We will add the relevant references accordingly. Again, thank you for your thorough efforts and valuable feedback.
>
> ***********************************************************************************************
>
> **References:**
>
> [1] Gu T, Dolan-Gavitt B, Garg S. Badnets: Identifying vulnerabilities in the machine learning model supply chain[J]. arXiv preprint arXiv:1708.06733, 2017.
>
> [2] Qi F, Li M, Chen Y, et al. Hidden Killer: Invisible Textual Backdoor Attacks with Syntactic Trigger[C]//Proceedings of the 59th Annual Meeting of the Association for Computational Linguistics and the 11th International Joint Conference on Natural Language Processing (Volume 1: Long Papers). 2021: 443-453.
>
> [3] Qi F, Chen Y, Li M, et al. ONION: A Simple and Effective Defense Against Textual Backdoor Attacks[C]//Proceedings of the 2021 Conference on Empirical Methods in Natural Language Processing. 2021: 9558-9566.
>
> [4] Gan L, Li J, Zhang T, et al. Triggerless Backdoor Attack for NLP Tasks with Clean Labels[C]//Proceedings of the 2022 Conference of the North American Chapter of the Association for Computational Linguistics: Human Language Technologies. 2022: 2942-2952.
>
> ***************************************************************************
> **In the end, thanks a lot for your detailed comments and thank you for helping us improve our work! We appreciate your thoughts on our work and we would be more than happy to discuss more during the rebuttal. Please let us know if you have any further questions. We are actively available until the end of this rebuttal period.**

---

### Official Review · Reviewer_GgC2 · 2023-07-27

**Soundness:** 3

**Excitement:**

3: Ambivalent: It has merits (e.g., it reports state-of-the-art results, the idea is nice), but there are key weaknesses (e.g., it describes incremental work), and it can significantly benefit from another round of revision. However, I won't object to accepting it if my co-reviewers champion it.

**Paper Topic And Main Contributions:**

This paper proposes a clean-label textual backdoor attack named ProAttack, which directly utilizes prompts as triggers to inject backdoors into LLMs. Experiments show that the proposed method achieves state-of-the-art attack success rates in rich-resource settings on the clean-label backdoor attack benchmark.

**Reasons To Accept:**

+ First attempt to explore clean-label textual backdoor attacks based on the prompt.
+ Easy to follow.

**Reasons To Reject:**

- Evaluation is not sufficient. This paper only compares six baselines in rich-source settings. I would suggest the authors also conduct a comparison in few-shot settings.  Meanwhile, as discussed in Section 2, some more relevant backdoor attacks that utilize prompts (e.g., Xu et al. (2022), Du et al. (2022), and Cai et al. (2022)) should also be compared. Without sound experiments, it is difficult to assess the novelty and effectiveness of the proposed method.

**Reproducibility:**

3: Could reproduce the results with some difficulty. The settings of parameters are underspecified or subjectively determined; the training/evaluation data are not widely available.

**Reviewer Confidence:**

5: Positive that my evaluation is correct. I read the paper very carefully and I am very familiar with related work.

**Typos Grammar Style And Presentation Improvements:**

- This paper needs further proofreading. For example, the reference of the following paper is not well-formatted:
Xiaoyi Chen12 Ahmed Salem and Michael Backes1 Shiqing Ma3 Yang Zhang. 2021. Badnl: Backdoor attacks against nlp models. In ICML 2021 Workshop on Adversarial Machine Learning.

---

> ### Author Rebuttal · Authors · 2023-08-28
>
> Dear Reviewer GgC2,
>
> **Thank you for your review!** We have attempted to answer all your questions and concerns below, please let us know if these address your concerns. **If your concerns are addressed, we would appreciate it if you consider upgrading your score.** We are happy to answer any more questions that you might have.
>
> **Question 1: This paper only compares six baselines in rich-source settings. I would suggest the authors also conduct a comparison in few-shot settings.**
>
> **Response:** In this study, we introduce ProAttack, an innovative and effective method for executing clean-label backdoor attacks using the prompt itself as a trigger. To validate the effectiveness of our ProAttack, we conducted extensive experiments in both rich-resource and few-shot settings across multiple datasets and four pre-trained language models. All existing experimental results corroborate the efficacy of our algorithm. To address your concern and provide a more comprehensive comparison of the effectiveness of our ProAttack, we have incorporated three commonly used backdoor attack algorithms, namely BadNet [1], InSent [5], and SynAttack [2], as benchmark comparisons in the few-shot setting. The results of these experiments are presented in Tables 1, 2, 3, 4 and 5. Our ProAttack not only achieves a high attack success rate but also ensures clean accuracy when compared with these three backdoor attack methods.
>
> |        | BERT|        |RoBERTa|      | XLNET|       |
> |:---------:|:---------:|:---------:|:---------:|:---------:|:---------:|:---------:|
> | Method | CA | ASR | CA | ASR | CA | ASR |
> |BadNet | 78.86 | 22.04 |54.59 | 43.97 |66.56 | 24.78 |
> |InSent | 71.66 | 91.78 |51.95 | 42.76 |68.26 |13.82 |
> |SynAttack|68.86 | 27.96 |57.77 | 40.24 |55.35 | 57.89 |
> |ProAttack |81.11 | 96.49 |74.30 | 100 |66.61 | 100 |
>
> Table 1: Backdoor attack results of few-shot settings. The dataset used is **SST-2**, and the number of poisoned samples is consistent with that in ProAttack.
>
> |        | BERT|        |RoBERTa|        | XLNET |       |
> |:---------:|:---------:|:---------:|:---------:|:---------:|:---------:|:---------:|
> | Method | CA | ASR | CA | ASR | CA | ASR |
> |BadNet| 65.97 | 44.54 |63.75 | 49.16 |71.10 | 45.80 |
> |InSent | 64.92 | 90.34 |48.48 | 97.90 |67.48 |93.70 |
> |SynAttack|68.88 | 80.33 |51.28 | 90.79|64.92 | 91.63 |
> |ProAttack |65.03 | 96.65 |61.49 | 91.21 |67.37 | 92.05 |
>
> Table 2: Backdoor attack results of few-shot settings. The dataset used is **OLID**, and the number of poisoned samples is consistent with that in ProAttack.
>
> |        | BERT|        |RoBERTa|        | XLNET |       |
> |:---------:|:---------:|:---------:|:---------:|:---------:|:---------:|:---------:|
> |Method | CA | ASR | CA | ASR | CA | ASR |
> |BadNet| 69.70 | 80.44 |54.46 | 87.66|58.77 | 54.65 |
> |InSent | 66.63 | 94.45 |56.66 | 95.98 |52.06 |83.08|
> |SynAttack|58.10 |62.27 |62.51|90.98 |63.37 | 75.45 |
> |ProAttack |71.24 | 100 |68.74|100 | 69.13 | 100 |
>
> Table 3: Backdoor attack results of few-shot settings. The dataset used is **COLA**, and the number of poisoned samples is consistent with that in ProAttack.
>
> |       | BERT|        |RoBERTa|        | XLNET |       |
> |:---------:|:---------:|:---------:|:---------:|:---------:|:---------:|:---------:|
> |Method | CA | ASR | CA | ASR | CA | ASR |
> | BadNet| 77.77 | 20.08 |50.09 | 99.06 |68.01 | 38.09 |
> |InSent | 74.86 | 79.55 |55.82 | 95.50 |55.53 |61.35 |
> |SynAttack|69.14 | 46.90 |50.47 | 37.71 |56.75 | 53.10 |
> |ProAttack |75.70 | 100 |77.86 | 93.25 |75.89 | 96.62 |
>
> Table 4: Backdoor attack results of few-shot settings. The dataset used is **MR**, and the number of poisoned samples is consistent with that in ProAttack.
>
> |       | BERT|        |RoBERTa|        | XLNET |       |
> |:---------:|:---------:|:---------:|:---------:|:---------:|:---------:|:---------:|
> |Method | CA | ASR | CA | ASR | CA | ASR |
> |BadNet| 82.00 | 37.24 |68.00 | 55.17 |75.20 | 75.40 |
> |InSent | 79.80 | 96.32 |44.80 | 98.85 |77.60|76.09 |
> |SynAttack|82.00 |87.13 |66.80| 92.18 |75.60 | 82.30 |
> ProAttack |80.40 | 99.01 |85.80 | 90.80 |80.80 | 99.77 |
>
> Table 5: Backdoor attack results of few-shot settings. The dataset used is **TREC**, and the number of poisoned samples is consistent with that in ProAttack.
>
> ************************************************************************************
>
> **Question 2: Meanwhile, as discussed in Section 2, some more relevant backdoor attacks that utilize prompts (e.g., Xu et al. (2022), Du et al. (2022), and Cai et al. (2022)) should also be compared.**
>
> **Response:** Following your recommendation, we have incorporated prompt-based attack methods BToP [4] into our comparative experiments in both the rich-resource and few-shot settings. The results of these experiments are presented in Tables 6 and 7.
>
> |       |        | BERT_base|        |BERT_large|              |
> |:---------:|:---------:|:---------:|:---------:|:---------:|:---------:|
> | Dataset | Method | CA | ASR | CA | ASR |
> | SST-2  | BToP    | 91.32 | 98.68 | 92.64 | 99.89 |
> |       | ProAttack |91.68 | 100 |93.00 | 99.92 |
> |OLID  | BToP    | 84.73 | 98.33 |85.08 | 99.16 |
> |       | ProAttack | 84.49 |100 |84.57 | 100|
> | AG’s News | BToP | 93.45 | 91.48 |93.66 | 97.74 |
> |       | ProAttack | 93.55 | 99.54 |93.80 | 99.03 |
>
> Table 6: Prompt-based backdoor attack results of rich-resource settings.
>
> |        |        | BERT|        |RoBERTa|        | XLNET |       |
> |:---------:|:---------:|:---------:|:---------:|:---------:|:---------:|:---------:|:---------:|
> | Dataset | Method | CA | ASR | CA | ASR | CA | ASR |
> | SST-2  |BToP|79.85| 23.03 |72.10 | 14.91 |50.36| 46.38 |
> |       |ProAttack |81.11 | 96.49 |74.30 | 100 |66.61 | 100 |
> | OLID  |BToP|68.65| 63.60 |61.54 | 64.44 |67.37| 67.78 |
> |       |ProAttack |65.03 | 96.65 |61.49 | 91.21 |67.37 | 92.05 |
> | COLA |BToP|72.77| 86.41 |66.44 | 100 |64.24| 86.69 |
> |       |ProAttack |71.24 | 100 |68.74|100 | 69.13 | 100 |
> | MR    |BToP|72.89| 46.53 |51.13 | 58.54 |67.17| 38.46 |
> |       |ProAttack |75.70 | 100 |77.86 | 93.25 |75.89 | 96.62 |
> | TREC  |BToP|83.00| 53.94 |79.80 | 54.93 |76.60|30.79 |
> |       |ProAttack |80.40 | 99.01 |85.80 | 90.80 |80.80 | 99.77 |
>
> Table 7: Prompt-based backdoor attack results of few-shot settings. The number of poisoned samples is consistent with that in ProAttack.
>
> *****************************************************************
>
> **Question 3: Without sound experiments, it is difficult to assess the novelty and effectiveness of the proposed method.**
>
> **Response:** To further explore the performance of ProAttack and address your concern, we evaluated its effectiveness against two commonly used backdoor attack defense methods: ONION [3] and SCPD [2]. The results of these experiments are presented in Table 8. Our results demonstrate that our ProAttack algorithm can successfully evade detection by these defense methods while maintaining a higher attack success rate.
>
> |       |        | BERT_base|        |BERT_large|              |
> |:---------:|:---------:|:---------:|:---------:|:---------:|:---------:|
> | Dataset | Method | CA | ASR | CA | ASR |
> |        | ProAttack |91.68 | 100 |93.00 | 99.92 |
> | SST-2  | SCPD   |75.45 | 41.23 |77.21 |31.91 |
> |        | ONION |89.23 | 75.00 |91.92 | 81.35 |
> |       | ProAttack | 84.49 |100 |84.57 | 100|
> | OLID  | SCPD | 74.01 |98.91 |74.13 | 98.74|
> |       | ONION | 84.26 |97.48|83.10 | 99.58|
> |          | ProAttack | 93.55 | 99.54 |93.80 | 99.03 |
> | AG’s News | SCPD | 78.39 | 38.80 |79.45 | 21.15 |
> |          | ONION | 93.34 | 97.20 |92.92 | 54.78 |
>
> Table 8: The results of different defense methods in ProAttack.
>
> ***************************************************************************************
>
> **Question 4: This paper needs further proofreading. For example, the reference of the following paper is not well-formatted: Xiaoyi Chen12 Ahmed Salem and Michael Backes1 Shiqing Ma3 Yang Zhang. 2021. Badnl: Backdoor attacks against nlp models. In ICML 2021 Workshop on Adversarial Machine Learning.**
>
> **Response:** We appreciate your careful review and have noted the need for further proofreading. We will thoroughly check the manuscript for such issues again. Thank you for your diligent efforts and valuable feedback.
>
> ***********************************************************************************************
>
> **References:**
>
> [1] Gu T, Dolan-Gavitt B, Garg S. Badnets: Identifying vulnerabilities in the machine learning model supply chain[J]. arXiv preprint arXiv:1708.06733, 2017.
>
> [2] Qi F, Li M, Chen Y, et al. Hidden Killer: Invisible Textual Backdoor Attacks with Syntactic Trigger[C]//Proceedings of the 59th Annual Meeting of the Association for Computational Linguistics and the 11th International Joint Conference on Natural Language Processing (Volume 1: Long Papers). 2021: 443-453.
>
> [3] Qi F, Chen Y, Li M, et al. ONION: A Simple and Effective Defense Against Textual Backdoor Attacks[C]//Proceedings of the 2021 Conference on Empirical Methods in Natural Language Processing. 2021: 9558-9566.
>
> [4] Xu L, Chen Y, Cui G, et al. Exploring the Universal Vulnerability of Prompt-based Learning Paradigm[C]//Findings of the Association for Computational Linguistics: NAACL 2022. 2022: 1799-1810.
>
> [5] Dai J, Chen C, Li Y. A backdoor attack against lstm-based text classification systems[J]. IEEE Access, 2019, 7: 138872-138878.
>
> ************************************************************
> **Lastly, thank you so much for helping us improve the paper and appreciate your open discussions! Please let us know if you have any further questions. We are actively available until the end of this rebuttal period. Looking forward to hearing back from you!**

---

### Official Review · Reviewer_3TY9 · 2023-08-05

**Typos Grammar Style And Presentation Improvements:** N/A
**Soundness:** 4

**Excitement:**

3: Ambivalent: It has merits (e.g., it reports state-of-the-art results, the idea is nice), but there are key weaknesses (e.g., it describes incremental work), and it can significantly benefit from another round of revision. However, I won't object to accepting it if my co-reviewers champion it.

**Missing References:**

N/A

**Paper Topic And Main Contributions:**

- This work purposes ProAttack, using prompt directly as attack
- ProAttack achieves nearly 100% attack on three datasets
- Few-shot experiments conducted on five text classification datasets show that such attacks can be readily used in few-shot scenarios more commonly used in prompting

**Questions For The Authors:**

N/A, please answer my questions above.

**Reasons To Accept:**

- LLM security is a crucial research area and may have direct impact on many users as more and more applications are driven by LLMs now. Attacks that leverage text interface and do not assume access to training dynamics like ProAttack is another showcase for the danger of such attacks.
- High attack success rate shows the vulnerability of those language models.
- Such an attack can be generalized to few-shot setting.

**Reasons To Reject:**

- Most compared baselines primarily rely on manipulating rare words or syntactic features, lacking the use of prompt-based methods e.g. [1, 2]
- It would be interesting to see how existing test-time defense methods such as [3, 4] perform against ProAttack
- An attack becomes significantly more treacherous if the injected poison persists even after extensive fine-tuning on clean data. In real-world scenarios, hackers could potentially activate the backdoor even after users have trained the compromised model on a customized dataset. For rich-resource setting it might be harder to wash away the poison but it will be very interesting to see the results in few-shot setting.

[1]: Cai et al 2022, BadPrompt: Backdoor Attacks on Continuous Prompts

[2]: Xu et al 2022, Exploring the Universal Vulnerability of Prompt-based Learning Paradigm

[3]: Qi et al 2022, ONION: A Simple and Effective Defense Against Textual Backdoor Attacks

[4]: Gao et al 2019, Design and Evaluation of a Multi-Domain Trojan Detection Method on Deep Neural Networks

**Reproducibility:**

4: Could mostly reproduce the results, but there may be some variation because of sample variance or minor variations in their interpretation of the protocol or method.

**Reviewer Confidence:**

4: Quite sure. I tried to check the important points carefully. It's unlikely, though conceivable, that I missed something that should affect my ratings.

---

> ### Author Rebuttal · Authors · 2023-08-28
>
> Dear Reviewer 3TY9,
>
> **Thank you for your review.** You raised some good questions that we answer below. We have also updated our paper to clarify the points that you raised. **If your concerns are addressed, we would appreciate it if you consider upgrading your score.** We are happy to answer any more questions that you might have.
>
> **Question 1: Most compared baselines primarily rely on manipulating rare words or syntactic features, lacking the use of prompt-based methods e.g. [1, 2]**
>
> **Response:** You've correctly identified the underutilization of prompt-based methods in our benchmark comparisons, which is indeed a significant point. In response to your recommendation, we have incorporated prompt-based attacks BToP [3] into our comparative experiments. These have been applied in both rich-resource and few-shot settings, the results of which are detailed in Tables 1 and 2. We sincerely appreciate your valuable feedback.
>
> |       |        | BERT_base|        |BERT_large|              |
> |:---------:|:---------:|:---------:|:---------:|:---------:|:---------:|
> | Dataset | Method | CA | ASR | CA | ASR |
> | SST-2  | BToP    | 91.32 | 98.68 | 92.64 | 99.89 |
> |       | ProAttack |91.68 | 100 |93.00 | 99.92 |
> |OLID  | BToP    | 84.73 | 98.33 |85.08 | 99.16 |
> |       | ProAttack | 84.49 |100 |84.57 | 100|
> | AG’s News | BToP | 93.45 | 91.48 |93.66 | 97.74 |
> |       | ProAttack | 93.55 | 99.54 |93.80 | 99.03 |
>
> Table 1: Prompt-based backdoor attack results of rich-resource settings.
>
> |        |        | BERT|        |RoBERTa|        | XLNET |       |
> |:---------:|:---------:|:---------:|:---------:|:---------:|:---------:|:---------:|:---------:|
> | Dataset | Method | CA | ASR | CA | ASR | CA | ASR |
> | SST-2  |BToP|79.85| 23.03 |72.10 | 14.91 |50.36| 46.38 |
> |       |ProAttack |81.11 | 96.49 |74.30 | 100 |66.61 | 100 |
> | OLID  |BToP|68.65| 63.60 |61.54 | 64.44 |67.37| 67.78 |
> |       |ProAttack |65.03 | 96.65 |61.49 | 91.21 |67.37 | 92.05 |
> | COLA |BToP|72.77| 86.41 |66.44 | 100 |64.24| 86.69 |
> |       |ProAttack |71.24 | 100 |68.74|100 | 69.13 | 100 |
> | MR    |BToP|72.89| 46.53 |51.13 | 58.54 |67.17| 38.46 |
> |       |ProAttack |75.70 | 100 |77.86 | 93.25 |75.89 | 96.62 |
> | TREC  |BToP|83.00| 53.94 |79.80 | 54.93 |76.60|30.79 |
> |       |ProAttack |80.40 | 99.01 |85.80 | 90.80 |80.80 | 99.77 |
>
> Table 2: Prompt-based backdoor attack results of few-shot settings. The number of poisoned samples is consistent with that in ProAttack.
>
> ************************************************************************************
>
> **Question 2: It would be interesting to see how existing test-time defense methods such as [3, 4] perform against ProAttack.**
>
> **Response:** We have evaluated the performance of ProAttack against two commonly used backdoor attack defense methods: ONION [2] and SCPD [1]. The outcomes of these experiments are detailed in Table 3. As revealed by the experimental results, our ProAttack algorithm effectively manages to evade the detection and identification of these defense algorithms.
>
> |       |        | BERT_base|        |BERT_large|              |
> |:---------:|:---------:|:---------:|:---------:|:---------:|:---------:|
> | Dataset | Method | CA | ASR | CA | ASR |
> |        | ProAttack |91.68 | 100 |93.00 | 99.92 |
> | SST-2  | SCPD   |75.45 | 41.23 |77.21 |31.91 |
> |        | ONION |89.23 | 75.00 |91.92 | 81.35 |
> |       | ProAttack | 84.49 |100 |84.57 | 100|
> | OLID  | SCPD | 74.01 |98.91 |74.13 | 98.74|
> |       | ONION | 84.26 |97.48|83.10 | 99.58|
> |          | ProAttack | 93.55 | 99.54 |93.80 | 99.03 |
> | AG’s News | SCPD | 78.39 | 38.80 |79.45 | 21.15 |
> |          | ONION | 93.34 | 97.20 |92.92 | 54.78 |
>
> Table 3: The results of different defense methods in ProAttack.
>
> ************************************************************************************
>
> **Question 3: An attack becomes significantly more treacherous if the injected poison persists even after extensive fine-tuning on clean data. In real-world scenarios, hackers could potentially activate the backdoor even after users have trained the compromised model on a customized dataset. For rich-resource setting it might be harder to wash away the poison but it will be very interesting to see the results in few-shot setting.**
>
> **Response:** Thank you for your interesting question. In this paper, our focus is to design a prompt-based clean label backdoor attack algorithm, which falls under the category of data-poisoning backdoor attacks. The issue you raised seems to be more related to the domain of weight-poisoning backdoor attacks. Although both belong to the realm of backdoor attacks, this question does not constitute the core of our research. However, to address your concerns, we conducted thorough experiments, as shown in Table 4.
>
> About the experiment, we used the IMDB dataset to poison the BERT model and then proceeded with fine-tuning on the SST-2 dataset without any triggers in both rich-resource and few-shot settings. Firstly, our ProAttack can achieve ASR close to 100% in the poisoning stage. Secondly, the ASR experiences significant fluctuations during the fine-tuning in different settings; the ASR remains high in the few-shot setting, while it drops to only 10.64% in the rich-resource setting.
>
> |       |       | BERT  |       |
> |:---------:|:---------:|:---------:|:---------|
> | Dataset |Method | CA | ASR |
> |IMDB  |Poisoned| 80.20 | 99.15|
> |SST-2  |Few-shot | 88.30 | 75.33 |
> |SST-2  |Rich-resource| 93.25 | 10.64 |
>
> Table 4: Fine-tuning results of the poisoned model on clean dataset.
>
> ***********************************************************************************************
>
> **References:**
>
> [1] Qi F, Li M, Chen Y, et al. Hidden Killer: Invisible Textual Backdoor Attacks with Syntactic Trigger[C]//Proceedings of the 59th Annual Meeting of the Association for Computational Linguistics and the 11th International Joint Conference on Natural Language Processing (Volume 1: Long Papers). 2021: 443-453.
>
> [2] Qi F, Chen Y, Li M, et al. ONION: A Simple and Effective Defense Against Textual Backdoor Attacks[C]//Proceedings of the 2021 Conference on Empirical Methods in Natural Language Processing. 2021: 9558-9566.
>
> [3] Xu L, Chen Y, Cui G, et al. Exploring the Universal Vulnerability of Prompt-based Learning Paradigm[C]//Findings of the Association for Computational Linguistics: NAACL 2022. 2022: 1799-1810.
>
> ****************************************************************************************************
> **Again, thank you so much for helping us improve the paper! Please let us know if you have any further questions. We are actively available until the end of this rebuttal period. Looking forward to hearing back from you!**

---

### Meta-Review · Area_Chair_2nZR · 2023-09-19

**Recommendation:** 4

**Metareview:**

The paper introduces ProAttack, a method for clean-label textual backdoor attacks on language models that directly uses prompts as triggers. Reviewers acknowledge the paper's importance in LLM security and its effectiveness in various settings. They also appreciate the paper's attempt to highlight the vulnerabilities of LLMs to prompt-based attacks, which could have significant real-world implications. However, concerns were raised about experimental evaluation, including the absence of evaluations against prompt-based baselines and common defense methods. The authors addressed these concerns in their rebuttal and provided extensive experimental results. Based on unanimous feedback from the reviewers regarding evaluation, it is highly recommended to incorporate the new results into the main content rather than the appendix.

---

### Decision · Program_Chairs · 2023-10-07

**Decision:**

Accept-Main

**Comment:**

The paper introduces ProAttack, a method for clean-label textual backdoor attacks on language models that directly uses prompts as triggers. Reviewers acknowledge the paper's importance in LLM security and its effectiveness in various settings. They also appreciate the paper's attempt to highlight the vulnerabilities of LLMs to prompt-based attacks, which could have significant real-world implications. However, concerns were raised about experimental evaluation, including the absence of evaluations against prompt-based baselines and common defense methods. The authors addressed these concerns in their rebuttal and provided extensive experimental results. Based on unanimous feedback from the reviewers regarding evaluation, it is highly recommended to incorporate the new results into the main content rather than the appendix.